



# Composition of gaseous organic carbon during ECOCEM in Beirut, Lebanon: new observational constraints for VOC anthropogenic emission evaluation in the Middle East

Thérèse Salameh[1], Agnès Borbon[1,a], Charbel Afif[2], Stéphane Sauvage[3,4], Thierry Leonardis[3,4], Cécile Gaimoz[1], and Nadine Locoge[3,4]

[1] Laboratoire Interuniversitaire des Systèmes Atmosphériques (LISA), IPSL, CNRS – UMR7583, University of Paris Est Créteil (UPEC) and Paris Diderot (UPD), Créteil, France

[2] Emissions, Measurements, and Modeling of the Atmosphere (EMMA) Laboratory, Unité Environnement, Génomique Fonctionnelle et Études Mathématiques, Centre d'Analyses et de Recherche, Faculty of Sciences, Saint Joseph University, Beirut, Lebanon

[3] Mines Douai, Sciences de l'Atmosphère et Génie de l'Environnement (SAGE), F-59508 Douai CEDEX, France

[4] Université de Lille, 59655 Villeneuve d'Ascq, France

[a] Now at : Laboratoire de Météorologie Physique (LaMP), CNRS - UMR 6016, University of Blaise Pascal, Clermont-Ferrand, France

**Abstract:**
The relative importance of Eastern Mediterranean emissions is suspected to be largely
underestimated compared to other regions worldwide. Here we use detailed speciated
measurements of VOCs (volatile organic compounds) to evaluate the spatial heterogeneity of
VOC urban emission composition and the consistency of regional and global emission
inventories downscaled to Lebanon (EMEP, ACCMIP, and MACCity). The assessment was
conducted through the comparison of the emission ratios extracted from the emission
inventories to the ones obtained from the hourly observations collected at a sub-urban site in
Beirut, Lebanon during summertime and wintertime ECOCEM campaigns. The observed ERs
were calculated by two independent methods. ER values from both methods agree very well
(difference up to 10%) and are comparable to the ones of the road transport sector from near-
field measurements for more than 80% of the species. There is no significant seasonality in ER
for more than 90% of the species unlike the seasonality usually observed in other cities
worldwide. Regardless of the season, ERs agree within a factor of 2 between Beirut and other
representative worldwide cities except for the unburned fuel fraction and ethane. ERs of
aromatics (benzene excepted) are higher in Beirut compared to northern post-industrialized
countries and even the Middle Eastern city Mecca. The comparison of the observed ER relative
to CO and to acetylene as well as the ratios of every NMVOC to each of the other NMVOCs
(NMVOC$_i$) to the ones extracted from ACCMIP and MACCity global emission inventories



suggests that the overall speciation of anthropogenic sources for major hydrocarbons that act
as ozone and SOA precursors in ACCMIP is reasonably represented.
The comparison of the specific road transport ER relative to acetylene derived from near-field
measurements to ER from ACCMIP and EMEP emission inventories for road transport sector
showed that ER from the road transport sector are usually consistent within a factor of 2 with
the regional emission inventory EMEP while xylenes and toluene are underestimated over a
factor of 2 by ACCMIP.
The observed heterogeneity of anthropogenic VOC emission composition between Middle
Eastern cities can be significant for reactive VOC but is not depicted by global emission
inventories. This suggests that systematic and detailed measurements are needed in the Eastern
Mediterranean Basin in order to better constrain emission inventory.

**Keywords:** emission inventory; sources; emission ratios; photochemical age; VOC; transport; East Mediterranean

**Highlights:**

- No significant seasonality in ER regardless of the species
- The transport sector is the major source of 80% of the species
- Reasonable discrepancies between ER from observations and global inventories
- Consistency of ER from the road transport sector within a factor of 2 with EMEP but underestimations of xylenes and toluene over a factor of 2 by ACCMIP
- Spatial heterogeneity of VOC emission composition in MEA region especially for reactive species

**1. Introduction**
In the context of global climate change and growing urbanization, the East Mediterranean Basin
(EMB) and the Middle East as a whole is a highly sensitive environment under considerable
anthropogenic and environmental pressures. Future decadal projections point to the EMB as a
possible hot spot of poor air quality and predict a continual and gradual warming in the region,
much stronger than other regions (Lelieveld et al., 2012, 2014; Pozzer et al., 2012). Future
projections using the Multi Pollutant Index defined by Gurjar et al. (2008) recently identified
North Africa and the Middle East as possible future hot spot of poor air quality (Pozzer al.,
2012). The increase and accumulation of anthropogenic emissions of gaseous and particulate
pollutants from surrounding urban areas, and on-road transport emissions in particular, are





suspected as one of the key compounding factors of those environmental impacts (Konovalov
et al., 2010; von Schneidemesser et al., 2011; Waked and Afif, 2012; Hillboll et al., 2013). The
EMB area includes two megacities: Istanbul (>12 million inhabitants) and Cairo (>15 million
inhabitants), which experience extremely high levels of pollutions (Kanakidou et al., 2011).
Satellite images of $NO_2$ columns from SCIAMACHY also point to the coastal urban areas of
the Middle East as an important hot spot of pollution (Lelieveld et al., 2009).
Trend analysis from satellite and ground-based observations found decreasing trend of primary
pollutants in Western Europe and increasing trends from hydrocarbon ground–based
observations and $NO_x$ satellite retrievals in other regions (Konovalov et al., 2010; von
Schneidemesser et al., 2011). Downward trends in pollutant emissions are a shared feature of
northern mid-latitude urban areas. The mixing ratios of VOCs (volatile organic compounds)
and CO have decreased by almost two orders of magnitude during the past five decades in Los
Angeles (Warneke et al., 2012). In UK, long-term trends show significant decreases for VOCs
reaching 26% per year as well as for CO up to 12% per year (von Schneidemesser et al., 2010).
Indeed, vehicle exhaust emission control has successfully reduced emissions of nitrogen oxides,
carbon monoxide, volatile organic compounds and particulate matter (Uherek et al., 2010). In
developing countries however, pollutants emissions have been growing strongly. Waked and
Afif (2012) have shown that the emissions of CO have rapidly increased by a factor of 2.8 in
the countries of the Middle East between 2000 and 2005, as a result of the increase of fuel
consumption. They also found that the road transport sector by the Middle East region is a
contributor to the global emissions of CO and $NO_x$ as significant as road transport by Western
Europe and North America. These findings are consistent with results reported by Uherek et al.
(2010) for a larger range of pollutants ($CO_2$, CO, $NO_x$, and NMVOCs).
Same picture can be drawn from emission inventories. Figure 1 compares the annual total
anthropogenic emissions of non-methane volatile organic compounds (NMVOCs) from three
reference emission inventories between the Middle East (MEA) region, Europe and North
America: ACCMIP (Atmospheric Chemistry and Climate Model Intercomparison Project),
EDGARv4.2 (Emissions Database for Global Atmospheric Research), and RCPs
(Representative Concentrations Pathways) (ECCAD database, 2015, http://eccad.sedoo.fr).
NMVOC emissions have been constantly increasing over the last thirty years reaching up to18
Tg/year according to RCPs 8.5 in 2010. On the opposite, the NMVOC emissions have been
strongly decreasing in USA and Europe, reaching respectively 7 and 10 Tg/year in 2010
according to RCPs 8.5 (Figure 1). While differences up to a factor of 2 can be found between
ACCMIP and EDGARv4.2 for the MEA in 2000 (Figure 1), all inventories suggest that




NMVOCs emissions from the MEA region are as significant as the ones from post-
industrialized regions or even higher. Finally no source regions clearly dominate global
anthropogenic emissions and therefore an accurate representation of anthropogenic emissions
in developing country regions like the MEA is of importance where highest uncertainties are
expected.
The quantification of emission distribution is a challenge, and even more in cities of the EMB
and the Middle East as a whole where local emission data are sparse. Indeed, emission
inventories usually combine bottom-up and top-down calculations to estimate emissions.
Comprehensive bottom-up calculations aggregate multiple local statistics on different emission
source categories where possible. Top-down calculations use regional or national activity data
and re-allocate emissions to finer scale by using spatial surrogates (e.g. population statistics at
the local level). Therefore, the uncertainties of numerous data sources are cumulated in the
overall estimation of emission amounts and along increasing scales (local to regional to global).
Granier et al. (2011) had assessed the evolution of anthropogenic and biomass burning
emissions of CO, $NO_x$, $SO_2$ and BC at global and regional scales from several inventories
during the 1980–2010 period and had concluded that there is still no consensus on the best
estimates for surface emissions of atmospheric compounds (Granier et al., 2011).
Several studies in the literature have reported evaluations of emission inventories in developed
countries by the use of ambient pollutant observations at ground level, on-board aircraft and
from satellite retrievals (Martin et al., 2003; Kim et al., 2011). For instance urban emission
ratios of various VOC relative to an inert tracer of incomplete combustion (CO, acetylene) have
been used as high-quality field constraints to evaluate regional emission inventories in cities of
post-industrialized countries (Warneke et al., 2007; Coll et al., 2010; Borbon et al., 2013). The
most recent diagnostic studies reveal large discrepancies between observations and inventories
as large as a factor 4 for VOC in Paris and Los Angeles for instance (Borbon et al., 2013). In-
situ observations are therefore necessary constraints for the development of reliable emission
inventories. Moreover, they also provide direct indications on air pollution exposure. While
some highly resolved inventories have been developed at the regional scale in the EMB area
for Beirut (Waked et al., 2012) and Istanbul (Im at al., 2011; Markakis et al., 2012), their
uncertainties are unknown and speciation of NMVOC is usually disregarded. For solely
Lebanon, the atmospheric emission inventory of anthropogenic and biogenic sources was
established by Waked et al. (2012) for a base year of 2010 for CO, $NO_x$, $SO_2$, total NMVOCs,
$NH_3$, $PM_{10}$ and $PM_{2.5}$, according to EEA/EMEP guidelines (EMEP/EEA, 2009). The NMVOC



total emissions in Lebanon were estimated to be 115 Gg for the year 2010. According to this
inventory, transport is the main source of NMVOC with a relative contribution of 67% of total
emissions of NMVOCs (Waked et al., 2012). In addition, there is a strong need for developing
better emission inventories in the Middle East region at a whole (Waked et al., 2013a). The
paucity of observations in this region, especially for VOCs and PM composition, is a strong
limitation to the achievement of evaluated and accurate emission inventories. Finally, accurate
modelling of individual NMVOCs and better understanding of ozone precursors is also
important for policy-makers and for the improvement of air quality leading to better estimates
of SOA (Secondary Organic Aerosols) formation and ozone levels.
The present paper aims at evaluating global VOC speciated emission inventories, in the absence
of regional emission inventories for MEA, by the use of detailed observations recently collected
in Beirut, Lebanon in the frame of ECOCEM project (Salameh et al., 2015). Those observations
include detailed near-source field measurements and ambient measurements at a suburban site.
We used regional (EMEP) and global (ACCMIP and MACCity) emission inventories
downscaled to Lebanon. Two independent methods already applied to Los Angeles basin during
the CalNex experiment (Borbon et al., 2013) that take into consideration the effects of chemical
removal were used to estimate the urban enhancement emission ratios (ERs) of various
NMVOC relative to CO and acetylene for winter and summer. First, calculated ERs are
compared to available ERs in other cities worldwide in order to assess the spatial variability of
emission composition. Second, calculated ERs are compared to the ones derived from global
emission inventories for all the anthropogenic sectors and for the road transport sector. Finally
perspectives for the whole Middle East region are discussed.
**2. Experiment**
The experimental strategy includes near field measurements close to major emission sources
and two intensive field campaigns conducted in summer 2011 and in winter 2012 in Beirut at a
suburban site, within the ECOCEM project (Emissions and Chemistry of Organic Carbon in
the East Mediterranean). VOC have been measured by a combination of on-line and off-line
techniques (table 1). A set of eight speciated profiles of four major non-methane hydrocarbons
(NMHC) sources in Lebanon was proposed by Salameh et al. (2014) by performing sampling
experiments close to emission sources in real-condition operation as far as possible. Field
sampling has been carried out in Beirut city and in the suburban area during March and April
2012. The sampling was performed by canisters and the analyses were performed in the



laboratory by thermal desorption-gas chromatography (TD-GC) technique coupled to a flame
ionisation detector (FID) from Perkin–Elmer (Salameh et al., 2014; 2015).
The ambient field campaigns were conducted on the roof of the Faculty of Sciences building of
Saint Joseph University located in the eastern suburbs of the city of Beirut from 2 to 18 July
2011 in summer and from 28 January to 12 February 2012 in winter. The site is appropriately
located in order to receive air masses coming from Greater Beirut Area which includes the city
of Beirut and close suburbs. The site is surrounded by a forested pine area and a high density
of residential premises. Beirut International Airport is located 8 km southwest of the site and
one fuel storage facility is located at 3.6 km North. More details are reported in Salameh et al.
(2015). During both measurement periods, NMHCs were continuously analyzed on an hourly
basis covering 30 min of ambient air sampling, by the same system (TD-GC-FID) used for
canister analysis. Additional measurements of trace gases concentrations including CO, $NO_x$
and $O_3$ were provided on a 1-min basis by specific analyzers. Basic meteorological parameters
(wind speed and direction, temperature, relative humidity and atmospheric pressure) were
measured on a 1-min basis for the duration of the campaigns.
During the summer field campaign, the average temperature was around 25 °C ± 2 °C and the
average wind speed was low, 2 m s$^{-1}$, with maximum wind speeds (4-10 m s$^{-1}$) recorded during
the days under south-western wind regimes and under northern wind regimes. During the winter
measurement campaign, the temperature stayed mild with an average of 13 °C ± 2 °C. The
average wind speed was still low at 2 m s$^{-1}$ and the wind direction was mostly south-easterly
and easterly.

**3. Regional and global emission database**
At the global and the regional scales, several emission inventories were developed during the
past few years in order to provide the distribution of surface emissions of different gaseous and
particulate compounds.
The EMEP regional emission inventory (European Monitoring and Evaluation Programme)
covers the period from 1980 to 2020, at a 0.5° resolution. It contains national total emissions
for Europe, sector data and gridded emission data for modelling purposes. The sectors follow
the SNAP categorization (Selected Nomenclature for reporting of Air Pollutants). Only the
anthropogenic part of the emissions regarding road transport (SNAP 07) is included in this
study.





Since there is no global emissions dataset available after 2000, the MACCity emissions have
been developed within two European Commission funded projects (MACC and CityZen) as an
extension of the ACCMIP and the RCP 8.5 emissions dataset. The emissions for each
compound were then linearly interpolated, for each sector and each year between 2000 and
2010. A seasonal cycle was applied for each sectoral layer, and the NMVOC species
anthropogenic emissions have been lumped to 13 species (excluding acetylene) using the same
methodology for the NMVOCs speciation as used in the ACCMIP dataset.
While a large number of emission inventories only provide the distribution of total NMVOC
emissions by lumping organic species (alkanes, alkenes, alkynes, alcohols, aldehydes, ketones
and aromatic compounds) for chemistry modelling purposes (EDGARv4.2, ECLIPSE-GAINS-
4a). We only rely on the ACCMIP inventory from ECCAD database which provides a detailed
speciation of VOCs including acetylene. ACCMIP global emission inventory covers the
historical period (1850–2000) in decadal increments at a horizontal resolution of 0.5° in latitude
and longitude, based on the combination of the best global and regional available datasets.
ACCMIP was developed mainly to provide consistent gridded emissions of reactive gases and
aerosols for use in chemistry model simulations needed by climate models for the Climate
Model Intercomparison Program 5 (CMIP5) in support of the Intergovernmental Panel on
Climate Change (IPCC) Fifth Assessment report (AR5) (Lamarque et al., 2010). 40 regions and
12 sectors were used to combine the various sources within this inventory and a set of species
including CO, NOx, total and speciated NMVOCs is provided. Speciation of NMVOC
emissions is performed using the RETRO (REanalysis of the TROpospheric chemical
Composition) (Schultz et al., 2007) inventory and is kept constant for the whole historical
period because of the lack of additional information. For all NMVOC species of the RETRO
inventory, a factor was calculated only for the year 2000 by dividing the emission of each
individual species by the total NMVOC anthropogenic emissions for each sector in each grid
cell.
For comparison with observations, we extracted the emission fluxes (kg/m$^2$/s) of 11 individual
NMVOCs as well as CO of 8 grids covering Lebanon surface area (ECCAD database). Target
sectors cover all anthropogenic sectors and the transport sector for 2000. Since the flux of the
8 grids is homogeneous and doesn't show large discrepancies in the spatial variability for each
considered compound, we used a mean flux for each NMVOC. This being said, the approach
we adopted by using the emission ratios collected at the measurement site in sub-urban Beirut
is valid. In addition, the site is far enough from strong direct emissions (industrial, road





transport) since they can hide the emissions from distant sources, and it receives air masses
coming from Greater Beirut Area which includes the city of Beirut and close suburbs.
**4. Emission and chemistry of gaseous organic carbon in Beirut on a seasonal basis**
The intensity of the emissions, chemistry, and atmospheric dynamics varying throughout the
day can affect the diurnal profiles of NMVOC absolute mixing ratios and as a consequence the
values of urban enhancement emission ratios (ER) (see section 5). We report in figures 2 and 3
for summer and winter, respectively, the average diurnal normalized profiles to the midnight
value of some NMVOC relative to an inert tracer to examine the relative importance of these
processes (Borbon et al., 2013). Acetylene was chosen as a relatively inert tracer and its
normalized diurnal profile is reported in each panel (grey shaded). The NMVOCs extremely
high concentrations have been filtered by removing the northern wind direction events,
referring to the identified gasoline evaporation episodic PMF (Positive Matrix Factorization)
factor (Salameh et al. 2016), in order to minimize the impact of the local industrial source
located in the northern part of the site (Salameh et al., 2015, 2016).
In summer all NMHCs including acetylene show the same bi-modal pattern with one midday
and one midnight maximum as a result of primary emissions from Greater Beirut transported
at the background receptor site between 09:00 and 10:00 and accumulation of emissions at night
in a shallower boundary layer. The degree of chemical removal during the day is $k_{OH}$-dependent
and increases with $k_{OH}$ as illustrated for alkenes and aromatics. At night alkenes and aromatics
follows the pattern of acetylene due to common sources, dilution and neglected photochemistry.
Then the magnitude of daytime maximum and minimum becomes modulated by chemical
removal when $k_{OH}$ is higher than $8.52 \times 10^{-12}$ cm$^3$ molecule$^{-1}$ s$^{-1}$. Compensation of chemical
removal at midday by evaporative emission cannot be excluded for aromatics like toluene.
In winter the diurnal profiles of NMHCs including acetylene show a unimodal pattern
characterized by maximum concentrations in the morning typical of traffic rush hours and a
slight decrease until late evening. Contrary to summer, the absence of a midday minimum of
normalized concentrations indicates a strong accumulation of primary emissions during the day
which is favored by poor dilution and negligible photochemistry. However, the diurnal profiles
of alkenes illustrated in figure 3 are characterized by a most pronounced enhancement in the
evening which can be related to additional combustion sources like domestic heating since the
measurement site is surrounded by a residential area (Salameh et al., 2015). Aromatics and
alkanes diurnal profiles follow the shape of acetylene profile in general. Nevertheless, one can



see a strong enrichment of aromatics during the day originating from the traffic related source
where toluene and m,p-xylenes are significantly present (Salameh et al., 2016).
In this section, we showed that some NMHC especially shorter-lived species like >C7-
aromatics, ethylene and propene could be affected by chemical removal in summer so that the
measured urban enhancement ratios can be different from the emission ratios. These results are
consistent with Salameh et al., 2015. The determination of ER in the next section will take into
account the potential effect of chemistry by applying temporal filters for the most reactive
species in summer. When NMHCs are not influenced by photochemistry, the enhancement
ratios will be estimated using all the data.
**5. Determination of urban enhancement emission ratios**
The emission ratio for NMVOC species is the ratio of a selected VOC to a reference compound
in fresh emissions without undergoing photochemical processing. We applied two methods to
determine emission ratios (ER) for each VOC species relative to CO and acetylene. The first
method consists on linear regression fits (LRF) on data applied to summer and winter datasets
and the other method relies on extrapolating the photochemical age to zero which was applied
to the summer data, photochemical removal being negligible in winter.
**5.1. Linear Regression Fit (LRF) method**
A commonly applied method to determine emission ratios is the linear regression fit (LRF) by
calculating the slope of the scatterplot between a particular NMVOC versus acetylene or CO
(Borbon et al., 2013). These reference compounds have been chosen due to their low reactivity
and since they are two tracers of incomplete combustion. Additionally, according to the national
emission inventory (Waked et al., 2012) and to PMF results (Salameh et al., 2016), the road
transport is the dominant source of NMVOCs in Lebanon and Greater Beirut; in urban areas as
well, emissions of CO are also dominated by vehicular emissions (von Schneidemesser et al.,

261    2010).

During both campaigns, as shown in the discussion in section 4, photochemical processing did
not play an important role in influencing NMVOC chemical compositions in ambient air of
Greater Beirut except for the most reactive species during the day in summer (alkenes and
aromatics). For the latter, emission ratios for NMVOC versus CO and for NMVOC versus
acetylene were derived from slopes of LRF for measurements made between 23:00 and 07:00
local time in summer when chemistry can be neglected . For isoprene, anthropogenic emissions
can be derived from nighttime data in the absence of its light-dependent biogenic origin (Borbon





et al., 2001). The LRF approach assumes that the composition of urban emissions relative to
CO and to acetylene does not change between day and night. For other species, the emission
ratios were derived from slopes of LRF for all the data in winter and summer since there is no
effect of photochemistry even during the day in summer, and in order to cover the urban mixing
of emissions from all the sources. The results are detailed in section 5.4 and summarized in
table 2.
**5.2. The Photochemical Age method**
The photochemical method takes into consideration the photochemical processing of measured
ratios of NMVOC with acetylene. We applied this method to all the summertime dataset.
Following de Gouw et al. (2005), Warneke et al. (2007), and Borbon et al. (2013), we assume
that the photochemical removal of NMVOC species is dominated by a reaction with the OH
radical. Firstly, the photochemical age of air masses, $\Delta t$, is estimated by using the NMVOC
ratios following this equation:
$$\Delta t = \frac{1}{[OH](k_{135TMB} - k_{benzene})} \times \left[ \ln\left( \frac{[135TMB]}{[benzene]} \right)_{t=0} - \ln\left( \frac{[135TMB]}{[benzene]} \right) \right]$$
(1)

Where [OH] is the concentration of hydroxyl radical, fixed to $5 \times 10^6$ molec.cm$^{-3}$; $k_{135TMB}$
($56.7 \times 10^{-12}$ cm$^3$.molec$^{-1}$.s$^{-1}$) and $k_{benzene}$ ($1.22 \times 10^{-12}$ cm$^3$.molec$^{-1}$.s$^{-1}$) are the reaction rate
coefficients with OH radical of 1,3,5-trimethylbenzene and benzene, respectively; [135TMB]
and [benzene] are the volume mixing ratios in ppb of benzene and 1,3,5-trimethylbenzene,
respectively. The 1,3,5-trimethylbenzene/benzene ratio was chosen to investigate the
photochemical age because these two compounds have similar sources but their lifetimes are
significantly different and lower than air mass transport timescale. The ratio of [135TMB] over
[benzene] at time zero (t=0) defines the emission ratio at a photochemical age of zero. This
emission ratio was derived from a scatter plot of [135TMB] vs. [benzene], it corresponds to the
emission enhancement ratio at nighttime data where photochemical processing can be
neglected.

294         Following de Gouw et al. (2005) and Warneke et al. (2007), the equations used for

emission ratio determination are described as follows for primary NMHCs (equation 2):
$$\frac{[NMVOC]}{[C_2H_2]} = ER_{NMVOC} \times \exp[-(k_{NMVOC} - k_{C_2H_2})[OH]\Delta t]$$
(2)





where [NMVOC] and [C$_2$H$_2$] are the volume mixing ratios in ppb of the NMVOC and acetylene
respectively; ER$_{NMVOC}$ is the emission ratio of the NMVOC relative to acetylene; k$_{NMVOC}$ and
k$_{C2H2}$ are the OH rate coefficients for the reaction of those compounds with OH radical (5.10$^6$
molecules.cm$^{-3}$); and $\Delta$t is the photochemical age calculated by equation 1. The emission ratios
of NMVOCs are estimated by extrapolating the photochemical age to zero. Following Borbon
et al. (2013) and Warneke et al. (2007), the emission ratios determined with the photochemical
age method are not affected when reducing or increasing the OH values by a factor of 2.
As an illustration, we report the ratios of measured benzene/acetylene, m,p-xylenes/acetylene,
and ethylene/acetylene as a function of the photochemical age in figure 4. Calculated
photochemical age is usually lower than 1h. As expected, the benzene/acetylene ratio keeps
constant with increasing photochemical age whereas a decrease of m,p-xylenes/acetylene and
ethylene/acetylene ratios is observed because they are more reactive. Surprisingly the
ethylene/acetylene ratio decreases faster than the (m,p)-xylenes/acetylene one while (m,p)-
xylenes are two times more reactive than ethylene. The decrease of the (m,p)-xylenes/acetylene
ratio might be modulated by additional evaporative emissions during the day that
counterbalance its chemical removal.

### 5.3. Performance of the different methods

Figure 5 illustrates the performance of the ER determination methods for the summertime data
where ER from the photochemical age and LRF methods are compared. There is an overall very
good agreement between the two methods (slope of the linear fit in red = 0.90 and correlation
coefficient of 0.99) showing the robustness of the methods and confirming that the selection of
all the data for less-reactive species and nighttime data for more reactive species for summer in
the LRF method does not bias the results. The calculated determination coefficients $R^2$ with
acetylene and CO ranged from 0.3 for some species like ethane to 0.9 for the majority of the
species in winter, and with acetylene in summer from 0.5 to 0.9. The standard deviation of the
ER determined by the photochemical method varied between zero and 0.04 for propane.

### 5.4. Emission ratios and seasonal variability

The ER derived from all wintertime and summertime (nighttime dataset for most reactive
species) datasets by the LRF method as well as the ER from the road transport established by
near-field measurements (Salameh et al., 2014) are summarized in Table 2. ER relative to CO
were not calculated in summer because CO data were not available for the whole campaign.





The ER derived from the observations (summer and winter) are comparable at ±50% to the
ratios of road transport sector for more than 80% of the species. This shows that urban emission
fingerprint is mainly driven by traffic emissions including ethane which does not have any
urban natural gas usage contribution in Lebanon. For other species like C4 to C6-alkenes,
styrene, and a few alkanes in winter the differences stay within a factor of 2, except for propane
where they are 5 in summer and 3 in winter. These differences can be partly explained by the
contribution of additional sources.
In general, the ERs in winter are slightly lower than in summer (Table 2). Figure 6 compares
the derived emission ratios of NMVOC species relative to acetylene at the measurement site
during summer and winter. The wintertime emission ratios for most NMVOC species agree at
±30% (slope of 0.71) and within a factor of 2 and a high determination coefficient of 0.94.
Finally, there is no significant seasonality in ER regardless of the species except for 1-pentene,
cyclohexane, styrene, 2,2-dimethylbutane, and 1,2,3-trimethylbenzene. Those species are
associated with ER lower than 0.1. These results indicate that the urban emission composition
contributing to the ambient concentrations is not significantly different between summer and
winter. The sources usually affected by seasonality are in winter related to combustion and to
fuel evaporation in summer. Our results based on ER are consistent with previous PMF results
(Salameh et al., 2016), showing that the road transport sector (combustion and gasoline
evaporation) is the dominating source in both seasons in Beirut.
These findings are different from the results usually found in recent studies. For instance,
Boynard et al. (2014) found that the emission ratios relative to acetylene in French cities (Paris
and Strasbourg) are slightly affected by the seasonality of emissions for benzene and alkenes
whereas other NMVOC species (alkanes and higher aromatics) exhibit 3 to 7 higher emission
ratios in summer. Moreover, Wang et al. (2014) reported lower wintertime emission ratios for
most NMVOC species than the summertime values by a factor ≥ 2 in Beijing, China. This
seasonal difference in emission ratios for most NMVOC species is possibly due to the seasonal
variations in NMVOC sources and in particular the modulation between wintertime combustion
and summertime evaporation.
**6. Comparison to other cities worldwide**
The NMVOC emission ratios relative to acetylene determined in Beirut by the LRF method are
compared to those recently determined in contrasted cities of North America (Los Angeles,
USA), Europe (Strasbourg, France), Middle East (Mecca, Saudi Arabia), and Asia (Beijing,
China,) on a seasonal basis in figure 7.



Usually ER agree within a factor of 2 except for aromatics (benzene excepted) and some alkanes
(C2 – C5). Those species are related to the unburned fuel fraction and natural gas or liquefied
petrol usage. Among C2-C5 alkanes and regardless of the season, ER of ethane is much lower
in Beirut than Los Angeles, Beijing, and Strasbourg but similar to Saudi Arabia since the natural
gas source is not widely used in Lebanon and Middle East countries (Salameh et al., 2015). The
maximum difference reached a factor of 10 for m,p-xylenes in winter. Regardless of the season,
ER of aromatics are higher in Beirut compared to northern post-industrialized countries and
even the Middle Eastern city Mecca. One should note that aromatic differences are quite
significant between the two Middle Eastern cities, from a factor of 3 up to a factor of 6 for
(m,p)-xylenes. ER of alkenes are higher in Mecca due to their additional evaporative origin
(Simpson et al., 2014). Differences are greater in winter than in summer as a consequence of a
marked seasonal variability of ER in opposite to Beirut. In Beirut, the aromatics are emitted
from combustion related sources and from gasoline evaporation which accounts for more than
40% in winter as well as in summer (Salameh et al., 2016). ER for alkenes, which are
combustion products, usually agree within a factor of two between Beirut, Los Angeles,
Beijing, and Strasbourg in both seasons, except for C4 and C5 alkenes with LA and Beijing.

## 7. Evaluation of global emission inventories for anthropogenic speciated VOC emissions

The emission fluxes (Kg/m$^2$/s) extracted from ECCAD database are converted to mole emission
fluxes, then the NMVOC ratios relative to CO and relative to acetylene are determined.
The comparison held here will consider the common compounds present in the ACCMIP and
MACCity, when possible, global emission inventories and measured during ECOCEM
campaigns which are listed in table 3. Important NMVOC species are present within these
inventories as tracers of many anthropogenic sources, reactive species, and important
precursors of ozone and SOA (Table 3).

### 7.1. Emission inventory vs. observations: all anthropogenic sectors

To analyze the consistency of the speciation of NMVOCs in the ACCMIP and MACCity global
emission inventories, we compared the individual NMVOC relative to CO as well as to
acetylene ratios during summer and winter. Figure 8 displays the ratios from the anthropogenic
emissions data (ACCMIP and MACCity) and the observations in a logarithmic scale for 11
individual NMVOC species color coded by the NMVOC groups. Except benzene, xylenes to a
less extent and long-lived alkanes, ER relative to acetylene agree within a factor of 2 between
observations and inventory suggesting that the overall VOC speciation in ACCMIP is



reasonably represented for more reactive VOCs. Regarding ER relative to CO, differences with
ACCMIP are remarkable. A global underestimation by the inventory by a shift towards lower
ER over an order of magnitude suggesting an overestimation of CO emissions by ACCMIP.

In order to consolidate our conclusions regarding VOC speciation within ACCMIP, we
performed the systematic calculation of the ratios of every NMVOC to each of the other
NMVOCs ($NMVOC_i$) in the global emission inventory and in the observations (Coll et al.,
2010). From figure 9, it appears that benzene is systematically overestimated up to a factor of
5 in ACCMIP and to a lesser extent, pentanes and butanes by a factor >2 whereas xylenes are
reasonably underestimated in the ACCMIP global emission inventory. The other compounds
lay around the line of the slope (= 1), below a factor of 2. Finally comparisons between
ACCMIP and observations (figures 8 and 9) suggest that the overall speciation of anthropogenic
sources for major hydrocarbons that act as ozone and SOA precursors in ACCMIP is reasonably
represented.
**7.2. Emission inventory vs. observations: road transport sector**
Our study has shown that calculated ER are comparable to the ones of road transport sector for
more than 80% of the species (table 2) which is consistent with PMF results (Salameh et al.,
2016). In Lebanon, PMF results showed that the major sources of NMHCs were traffic-related
emissions (combustion and gasoline evaporation) in winter and in summer accounting for 51
and 74 wt% respectively. Moreover, according to Parrish et al. (2009), the largest source of
emissions in most urban areas is road traffic, which includes tailpipe and evaporative emissions.
Therefore, it is also crucial to assess the emission inventories regarding the road transport sector
namely ACCMIP and EMEP SNAP 07.
We proceeded as in the previous section by comparing the road transport ER relative to
acetylene for all VOCs and individual VOCs from ACCMIP and EMEP emission inventories
to the ER from near-field measurements (figures 10 and 11) (Salameh et al., 2014). ER from
the road transport sector are usually consistent within a factor of 2 for the regional emission
inventory EMEP while xylenes and toluene are underestimated over a factor of 2 by ACCMIP.
At a more detailed level, by calculating the ratios for individual NMVOC, figure 11 confirms
that xylenes and toluene are underestimated species by both inventories.



### 7.3. Perspectives for Middle East region (MEA)


After a focus on Lebanon, the purpose of this last section is to provide some perspectives
regarding the whole MEA region. Figure 12 shows the comparison of ER relative to acetylene
from the ACCMIP emission inventory considering all the anthropogenic sectors of Lebanon
compared to the ones of four other Middle Eastern countries that are expected to be high VOC
emitters (Saudi Arabia, Egypt, Iran and Turkey). An overall homogeneity within a factor of 2
of the ERs is observed between countries of the Mediterranean border (Lebanon, Turkey, and
Egypt). However, when comparing Lebanon to other Middle East countries, some gaps are
depicted: The emission inventory suggests that the ERs of benzene, toluene, as well as of C4-
C5 alkanes are lower in Lebanon compared to Iran and Saudi Arabia. For the latter those
observations are completely contradictory with the ones reported in figure 7. We have shown
that the anthropogenic emissions in Beirut were more enriched in aromatics and propane and
poorer in alkenes than the ones of Mecca while benzene ER was consistent between both
countries. While the comparison here is limited by the number of species compared to figure 7
this suggests that the global emission inventory does not reproduce the heterogeneity of VOC
anthropogenic emission composition between countries of the MEA and same results can be
expected for all the MEA countries where emissions data and measurements are scarce.
Systematic and additional observations are needed in order to test the importance of such spatial
variability in anthropogenic VOC emission composition.

### 8. Conclusions


Detailed measurements of NMVOCs collected at a sub-urban site in Beirut, Lebanon, have been
used to evaluate regional and global emission inventories (ACCMIP, MACCity and EMEP)
downscaled to the studied domain. These data were collected during two intensive field
campaigns in summer 2011 and in winter 2012 within the framework of the ECOCEM project.
The emission ratios (ER) of individual NMVOC species relative to CO and acetylene were
successfully derived from ambient measurements in summer and in winter as well as from near-
field measurements for the road transport sector by applying two independent methods: the
linear regression fit method and the photochemical age method. Emission ratios from both
methods show a very good agreement at ±10%.
The ER derived from the observations (summer and winter) are comparable to the ratios of the
road transport sector for more than 80% of the species. There is generally no significant
seasonality in ER regardless of the species unlike the seasonality usually observed in other





cities. These results are consistent with the significant contribution of road transport sector
(combustion and gasoline evaporation) in winter and summer (Salameh et al., 2016).
Regardless of the season, ER derived from observations agree within a factor of 2 between
Beirut and other representative worldwide cities except for the unburned fuel fraction and
ethane. Aromatics (benzene excepted) show the largest differences up to a factor of 10 for m,p-
xylenes compared to northern post-industrialized countries and even another middle eastern
city like Mecca in Saudi Arabia.
ER relative to CO and to acetylene as well as the ratios of every NMVOC to each of the other
NMVOCs ($NMVOC_i$), extracted from ACCMIP and MACCity global emission inventory were
compared with the corresponding observed ER during both seasons, for all anthropogenic
sectors. This comparison suggests that the overall speciation of anthropogenic sources for major
hydrocarbons that act as ozone and SOA precursors in ACCMIP is reasonably represented.
The road transport ER relative to acetylene derived from near-field measurements are compared
to ER from ACCMIP and EMEP regional emission inventory for road transport sector. ER from
the road transport sector are usually consistent within a factor of 2 for the regional emission
inventory EMEP while xylenes and toluene are underestimated over a factor of 2 by ACCMIP.
It should be emphasized that when a consensus is met between observed and inventory ER (the
ER lay around the ratio of 1), this does not necessarily mean that the absolute emissions are
correct. Indeed, Salameh et al. (2016) have shown that global inventories (ACCMIP, EDGAR,
MACCity) could underestimate the NMVOC emissions up to a factor of 10 for the
transportation sector. Both speciation and absolute emissions have to be taken into
consideration.
Finally, we have shown that the emission inventory is in disagreement with the observations
when comparing Lebanon with Saudi Arabia. The observed heterogeneity of anthropogenic
VOC emission composition can be significant for reactive VOC (factor of 6 for m,p-xylenes)
but is not depicted by global emission inventories. This suggests that systematic and detailed
measurements are needed in the MEA region in order to better constrain emission inventory.
VOC emission inventory is the fundamental input of air quality modelling, therefore it plays a
major role in characterizing secondary pollution and control policy formulation. To improve
the quality of future VOC emission estimates, more efforts should be made toward refinement
of source classification, development of representative local emission factors, comprehensive
collection of activity data, and more accurate spatiotemporal characterization. Additionally,
comparison of available datasets will allow a quantification of the uncertainties on emissions.





At a regional and global level, long term and continuous studies integrating more than one
measuring site and more specific tracers are of great interest in order to provide more reliable
information and the use of surface observations from monitoring stations could help defining
better speciations.
**Acknowledgments :**
Funding for this study was obtained from Mines Douai Institution, the Lebanese National
Council for Scientific Research, Saint Joseph University (Faculty of Sciences and the Research
Council), CEDRE (Coopération pour l'Évaluation et le Développement de la Recherche) and
PICS no. 5630 (Programme Interorganismes de Coopération Scientifique du CNRS). This work
is also part of the ChArMEx program. ChArMEx is the atmospheric component of the French
multidisciplinary program MISTRALS (Mediterranean Integrated Studies aT Regional And
Local Scales). ChArMEx-France was principally funded by INSU, ADEME, ANR, CNES,
CTC (Corsica region), EU/FEDER, Météo-France and CEA. Thérèse Salameh's postdoc is
supported by DIM R2DS from Région Ile-de-France (2014-09).

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




**Table 1**: Measurements during ECOCEM Campaigns.

| Species | Instrument | Time resolution | Performances | References |
|---|---|---|---|---|
| **On-line**<br>67 C2-C9 NMHC<br>29 alkanes<br>19 alkenes<br>2 alkynes<br>17 aromatics | TD-GC-FID | 1 h | DL: 40 - 90 ppt, uncertainty: 4 - 35% for the majority of the NMHCs | Salameh et al., 2014, 2016 |
| **Off-line**<br>67 C2-C9 NMHC | Canister / TD-GC-FID | Sampling < 3 min / 1h analysis | DL: 40 - 90 ppt, uncertainty: 15% | Salameh et al., 2014, Sauvage et al., 2009 |

**Table 2:** Urban Emission Ratios of VOCs Relative to Acetylene and CO in Beirut in summer and in winter obtained by calculating a Linear Regression Fit to Data. The ERs from road transport by near-field measurements are also reported. Bold characters indicate the similarity at ±50% of the ER from the measurement campaigns to the one of near-field measurements.

| NMVOC | $\Delta VOC/\Delta C_2H_2$ (ppb/ppb) | | | $\Delta VOC/\Delta CO$ (ppb/ppm) |
| | Beirut – summer 2011 | Beirut – winter 2012 | road transport-near field spring 2012 measurement (Salameh et al., 2014) | Beirut – winter 2012 |
|---|---|---|---|---|
| ethane | **0.23** | **0.18** | **0.21** | 1.50 |
| propane | 1.02 | 0.58 | 0.21 | 5.40 |
| n-butane | **1.14** | **0.95** | 1.87 | 6.70 |
| i-butane | 0.25 | 0.48 | 0.72 | 3.30 |
| n-pentane | 0.26 | 0.15 | 0.40 | 1.10 |
| i-pentane | **1.22** | 0.75 | **1.77** | 5.30 |
| 2-methyl-pentane | **0.29** | 0.18 | **0.30** | 1.20 |
| 3-methyl-pentane | **0.17** | 0.11 | **0.16** | 0.80 |
| 2,2-dimethylbutane | **0.10** | 0.03 | **0.11** | 0.20 |
| n-hexane | **0.11** | **0.10** | **0.10** | 0.90 |
| Methylcyclopentane | 0.13 | 0.11 | 0.06 | 0.90 |
| 2-methyl-hexane | **0.11** | **0.08** | **0.08** | 0.70 |
| 3-methyl-hexane | **0.11** | **0.09** | **0.09** | 0.70 |
| cyclohexane | 0.04 | **0.01** | **0.01** | 0.10 |
| n-heptane | **0.07** | **0.05** | **0.06** | 0.40 |



| | | | | |
|---|---|---|---|---|
| 2,3-dimethylpentane | **0.04** | **0.03** | **0.03** | 0.20 |
| Methylcyclohexane | **0.04** | **0.03** | **0.03** | 0.20 |
| 3-methyl-heptane | **0.03** | **0.02** | **0.02** | 0.20 |
| n-octane | 0.04 | **0.02** | **0.02** | 0.20 |
| 2,2,4-trimethylpentane | **0.17** | **0.13** | **0.19** | 1.00 |
| 2,3,4-trimethylpentane | **0.06** | 0.04 | **0.08** | 0.30 |
| n-nonane | 0.04 | 0.03 | 0.01 | 0.20 |
| acetylene | - | - | - | 8.20 |
| propyne | **0.05** | **0.04** | **0.04** | 0.40 |
| ethene | 1.59 | **1.02** | **0.97** | 9.30 |
| propene | 0.47 | **0.32** | **0.31** | 3.00 |
| 1-butene | **0.10** | **0.09** | 0.20 | 0.70 |
| cis-2-butene | **0.13** | 0.08 | **0.15** | 0.60 |
| trans-2-butene | **0.16** | 0.11 | **0.19** | 0.90 |
| isobutene | 0.17 | 0.12 | 0.42 | 1.00 |
| 3-methyl-1-butene | **0.03** | 0.02 | **0.05** | 0.10 |
| 2-methyl-1-butene | 0.09 | 0.06 | 0.15 | 0.40 |
| 1-pentene | 0.05 | 0.17 | 0.34 | 1.40 |
| cis-2-pentene | 0.05 | 0.05 | 0.11 | 0.40 |
| trans-2-pentene | 0.11 | 0.11 | 0.25 | 0.80 |
| 1,3-butadiene | **0.08** | **0.05** | **0.07** | 0.50 |
| isoprene | **0.04** | 0.02 | **0.04** | 0.20 |
| Cyclopentene | **0.02** | 0.01 | **0.03** | 0.10 |
| Methylcyclopentene | **0.02** | 0.01 | **0.02** | 0.10 |
| 1-hexene | **0.02** | 0.01 | **0.03** | 0.10 |
| benzene | **0.25** | **0.23** | **0.24** | 2.00 |
| toluene | **1.56** | **1.33** | **1.09** | 11.10 |
| m,p-xylenes | **0.81** | **0.57** | **0.61** | 4.80 |
| o-xylene | 0.27 | **0.19** | **0.19** | 1.70 |
| Ethylbenzene | 0.23 | **0.16** | **0.15** | 1.40 |
| n-propylbenzene | **0.03** | **0.02** | **0.02** | 0.20 |
| Isopropylbenzene | **0.01** | **0.01** | **0.01** | 0.04 |
| m-ethyltoluene | 0.14 | **0.09** | **0.09** | 0.80 |
| p-ethyltoluene | **0.06** | **0.04** | **0.04** | 0.30 |
| o-ethyltoluene | **0.04** | **0.03** | **0.04** | 0.30 |
| styrene | 0.05 | **0.02** | **0.02** | 0.20 |
| 1,3,5-trimethylbenzene | 0.08 | **0.04** | **0.05** | 0.30 |
| 1,2,3-trimethylbenzene | **0.05** | 0.01 | **0.04** | 0.08 |
| 1,2,4-trimethylbenzene&decane | **0.24** | **0.16** | **0.19** | 1.40 |

nd: not determined




**Table 3**: List of target species and their $K_{OH}$ ((Atkinson and Arey 2003; Atkinson 2007) and $Y_{SOA}$ (SOA formation potential) ((Derwent et al., 2010)

| ACCMIP nomenclature MACCity* | Compounds considered from ECOCEM database | $K_{OH}$ $(10^{-12} cm^3.molécule^{-1}.s^{-1})$ | $Y_{SOA}$ |
|---|---|---|---|
| Ethane* | Ethane | 0.25 | 0.1 |
| Propane* | Propane | 1.09 | 0 |
| Butanes | Butane and isobutane | 2.36 and 2.12 | 0.3 and 0 |
| Pentanes | Pentane and isopentane | 3.8 and 3.6 | 0.3 and 0.2 |
| Ethene* | Ethene | 8.52 | 1.3 |
| Propene* | Propene | 26.3 | 1.6 |
| Acetylene | Acetylene | 0.9 | 0.1 |
| Benzene | Benzene | 1.22 | 92.9 |
| Trimethylbenzene | Sum of 1,2,3-1,2,4-,1,3,5-trimethylbenzene | 32.7; 32.5; 56.7 | 43.9; 20.6; 13.5 |
| Toluene | Toluene | 5.63 | 100 |
| Xylenes | Sum of m,p-xylenes and o-xylene | 13.6; 23.1; 14.3 | 95.5; 84.5; 67.1 |
| CO* | CO | | |

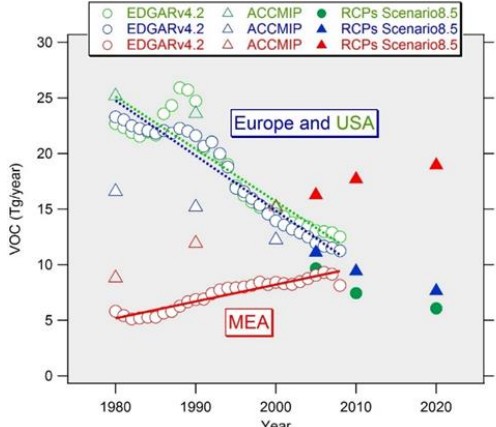

**Figure 1:** Evolution of the NMVOC anthropogenic total emissions (Tg/year) in the MEA, USA and Europe (OECD Europe and Eastern Europe) from 1980 to 2020 from reference global emission inventories (ECCAD database).



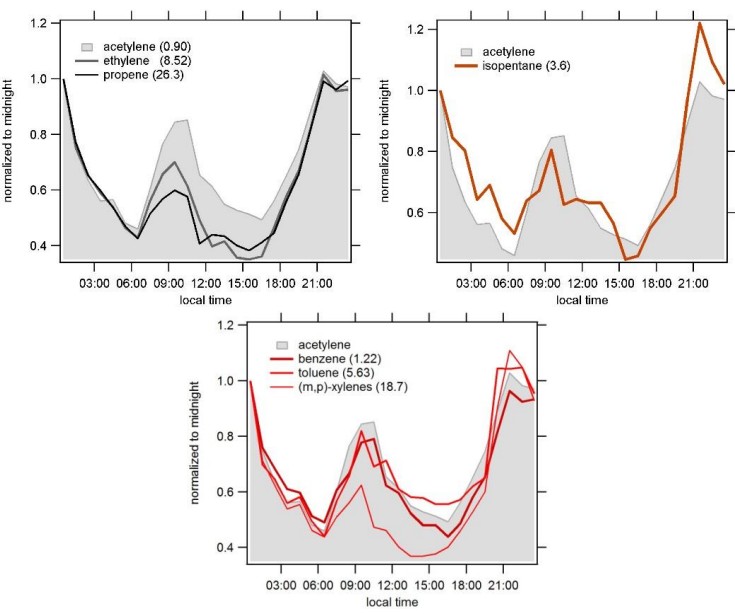

**Figure 2**: Normalized diurnal profiles of the mixing ratios of selected species to midnight values in summer. Numbers in parentheses are the rate coefficient with OH $\times 10^{-12}$ cm$^3$ molecule$^{-1}$ s$^{-1}$ (Atkinson and Arey 2003; Atkinson 2007).

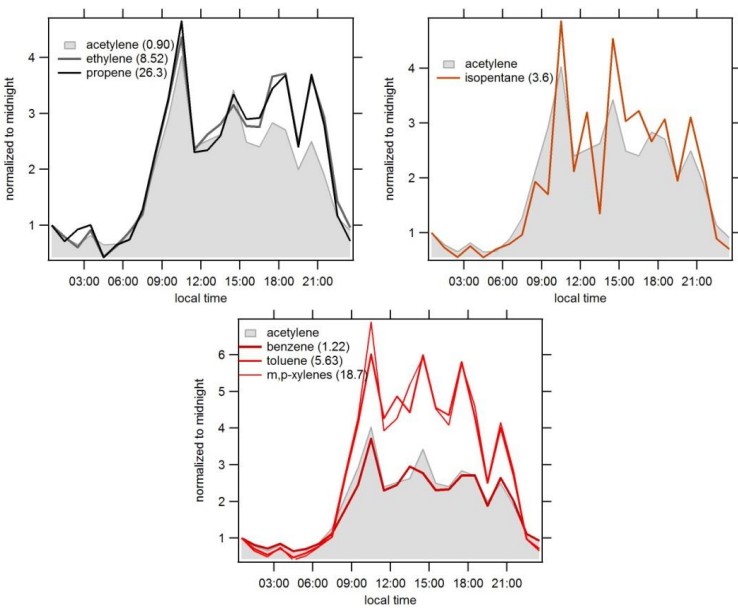

**Figure 3**: Normalized diurnal profiles of the mixing ratios of selected species to midnight values in winter. Numbers in parentheses are the rate coefficient with OH $\times 10^{-12}$ cm$^3$ molecule$^{-1}$ s$^{-1}$ (Atkinson and Arey 2003; Atkinson 2007).



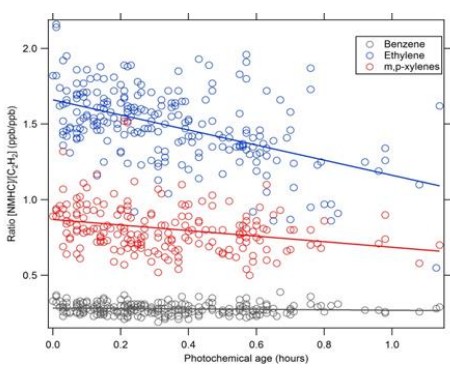

**Figure 4**: Measured ratios of benzene/acetylene, m,p- xylenes/acetylene, and ethylene/acetylene as a function of photochemical age (hours).

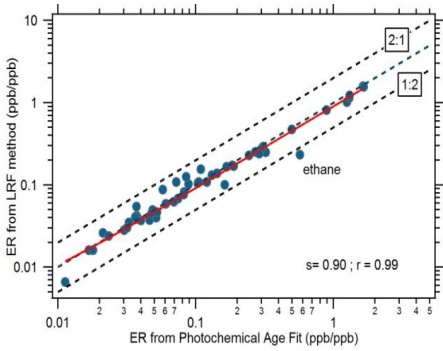

**Figure 5**: Comparison of the emission ratio (ER) estimated relative to acetylene by the nighttime linear regression fit and the photochemical age fit in summer.

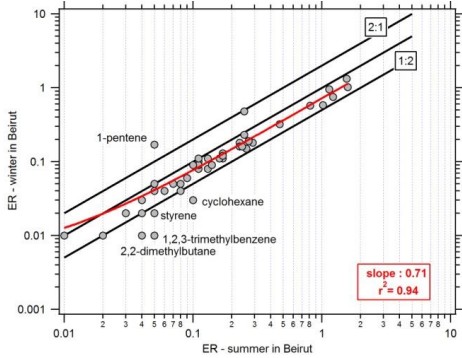

**Figure 6**: Comparison of summertime and wintertime emission ratios vs. acetylene in Beirut.





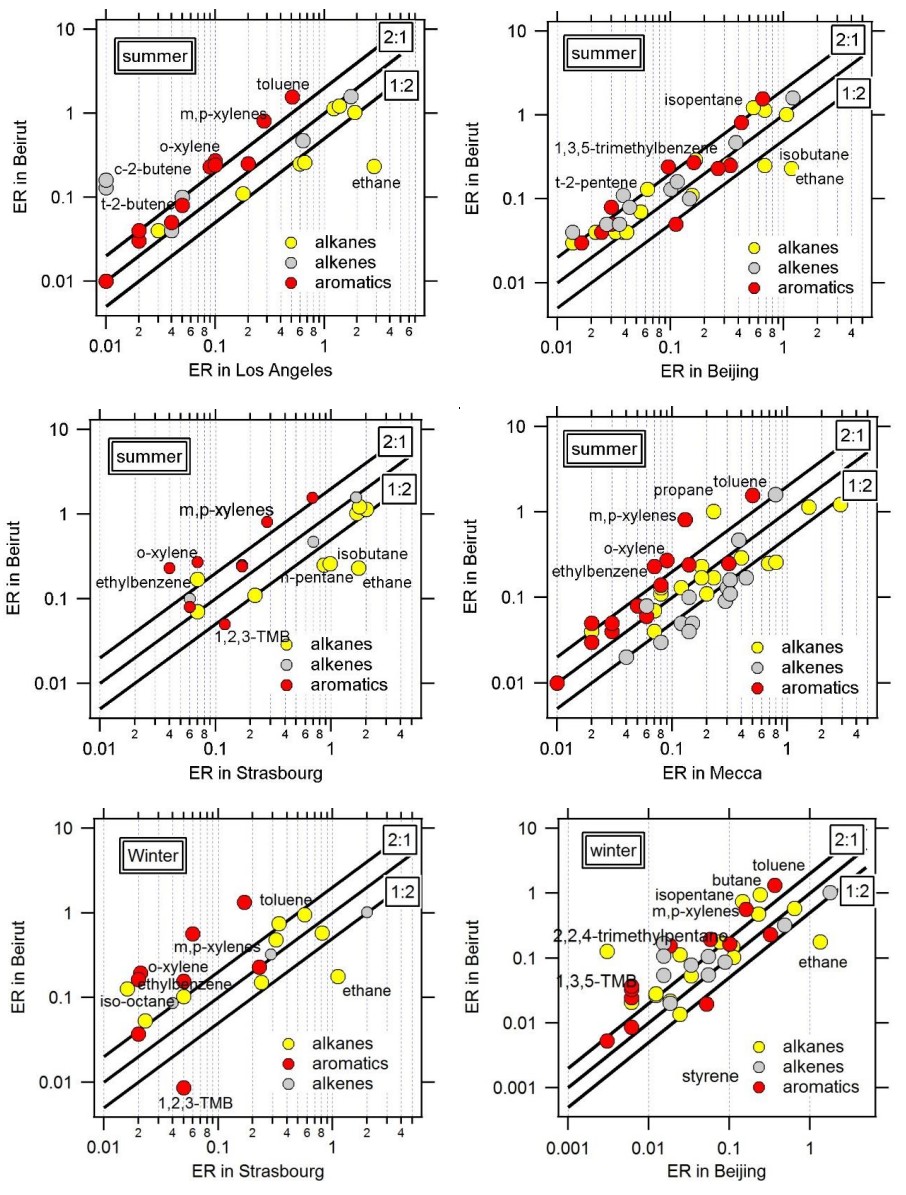

**Figure 7**: Comparisons of NMVOC emission ratios determined at the Beirut sub-urban site with those for Los Angeles, US (Borbon et al., 2013); Strasbourg, France (Boynard et al., 2014); Beijing, China (wang et al., 2014): and Mecca, Saudi Arabia (Simpson et al., 2014). (TMB: Trimethylbenzene)





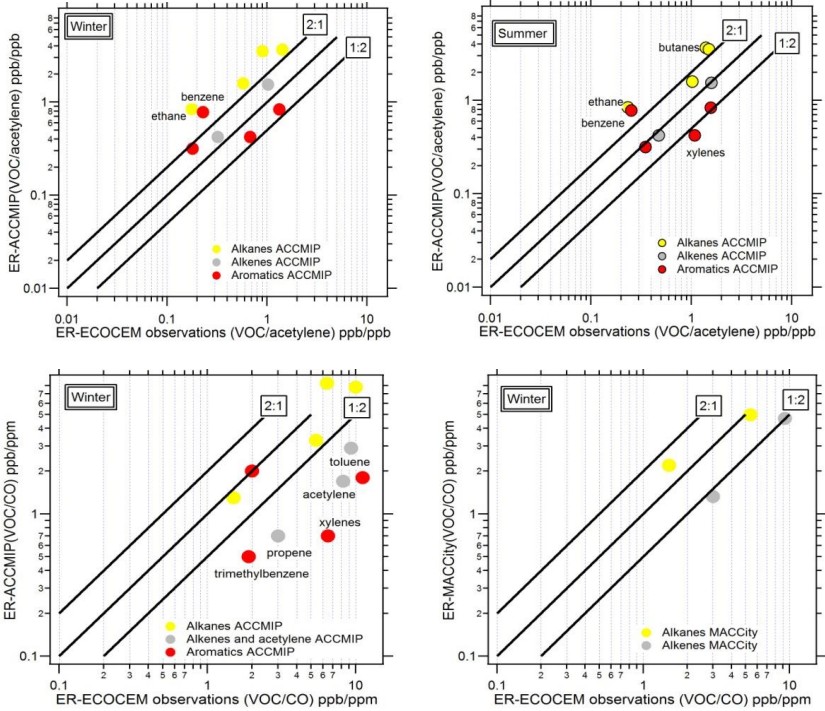

**Figure 8**: Comparison of the emission ratios from ACCMIP and MACCity to the measured ones by LRF method, in summer and in winter, relative to CO and acetylene, for all the anthropogenic sectors.



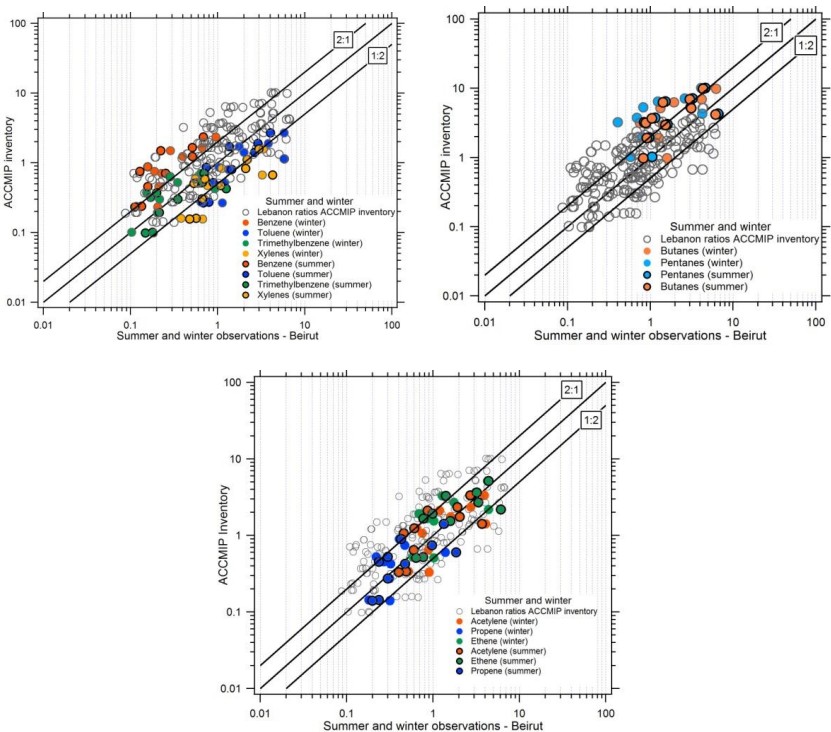

**Figure 9**: Comparison of the emission ratios of NMVOC vs. NMVOCi from ACCMIP to the measured ones, in summer and in winter, for all the anthropogenic sectors, for all data (in grey dots) and for a given NMVOC (colored dots).

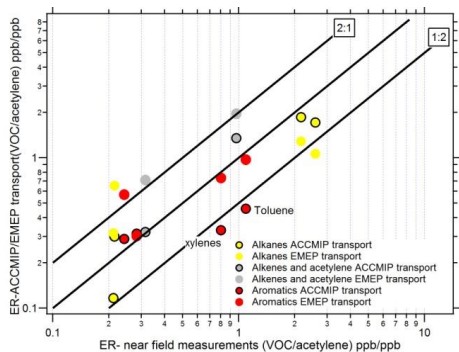

**Figure 10**: Comparison of the emission ratios relative to acetylene, from ACCMIP and EMEP emission inventories for road transport to the ER from road transport by near-field measurements.



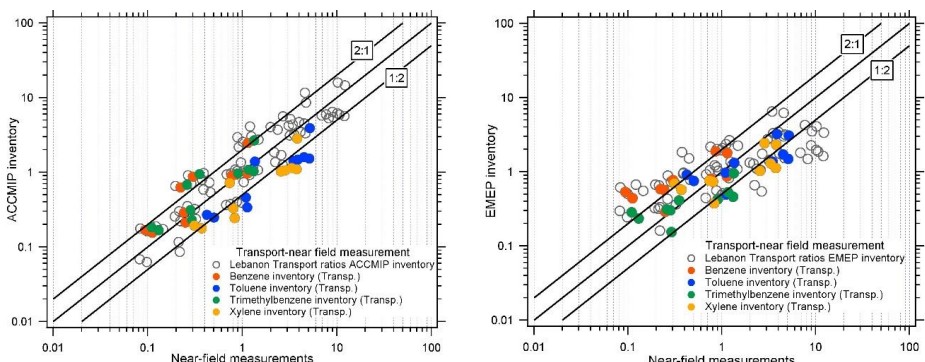

**Figure 11**: Comparison of the emission ratios of NMVOC vs. NMVOCi from ACCMIP and EMEP to the ER from near-field measurements for the road transport sector, for all data (in grey dots) and for a given NMVOC (colored dots).

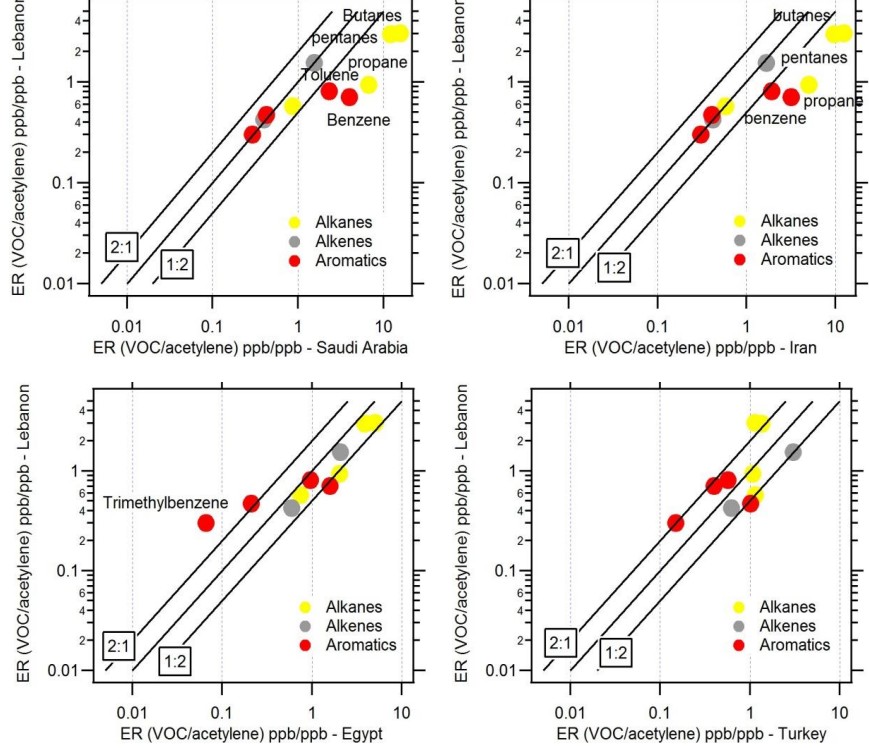

**Figure 12:** Comparison of the emission ratios from ACCMIP relative to acetylene for all the anthropogenic sectors for five Middle Eastern countries (Lebanon, Egypt, Turkey, Iran, and Saudi Arabia).