# Peer review of "Composition of gaseous organic carbon during ECOCEM in Beirut, Lebanon: new observational constraints for VOC anthropogenic emission evaluation in the Middle East"

_Atmospheric Chemistry and Physics, 2016_

## Referee Comment (RC1) · Anonymous Referee #1 · 8 Sep 2016

Review of 'Composition of gaseous organic carbon during ECOCEM in Beirut, Lebanon: new observational constraints for VOC anthropogenic emission evaluation in the Middle East'

ACPD Manuscript: This paper provides a nice presentation of emission ratios (ER) calculated by two different methods, and then follows up with a comparison with regional to global emission inventories. This paper is especially relevant given the lack of measurement data from this region of the world. Overall, the paper is well organized, and quite relevant. I would however like to see a few revisions before it is accepted for

publication. More general comments and specific edits/suggestions, including some language corrections, follow.

General comments:

-In general, the paper cites a lot of work from authors, which is fine given the lack of work in this region on the topic, but it would be good if there could also be some more citations that are beyond self-citation. For example, on L238, that toluene and xylenes are related to traffic sources, there are references other than Salameh 2016 that are earlier and more authoritative on this, as this is something that is more broadly true and not limited to the Middle East. It would be good to integrate some of these as well.

L95: CO is not inert. It is much less reactive than many of the VOCs but it is definitely not inert.

P7: this text discusses where speciations in emission inventories come from, but it does not make clear which species/speciation information is used from which sources for the comparisons in the study. Please try to be more explicit/clearer about this, since a variety of inventories, etc are discussed.

L202-204: This sentence does not make sense. Why does what is said above make the approach for Beirut valid? Please explain this more clearly.

L252: I find it very hard to believe that photochemical removal/photochemistry is truly 'negligible' in winter.

L270-274: Here the authors state that ER were derived using all the data 'since there is no effect of photochemistry even during the day in summer'. That simply cannot be true. No photochemistry during summer mid-day? If the authors wanted to argue that the LRF were derived based on all data because the emissions were fresh and therefore had not undergone significant photochemistry I could follow the argument/logic, but to state no photochemistry seems very unreasonable. Also, in section 4 the degree of chemical removal is stated to be kOH dependent. Overall, the discussion in section

4 and section 5 should be made more consistent, but if 'no photochemistry' rather than e.g., 'fresh emissions' wants to be argued, the data to prove this will need to be much more substantial. That there is substantial influence from emissions is discussed and a relevant point.

Section 5.2: It would be helpful to bring Figure 4 into the discussion earlier. I think visualizing this and referring to the plot will help the explanation. It would also be helpful if a number of points could be more explicitly mentioned. The ER is determined by extrapolating the photochemical age to t=0, so that means the y-intercept is the ER, correct? State this explicitly. Photochemical age is being shown on the x-axis, but what does that correspond to in terms of the equation that is shown? It would be good to be explicit about this too.

L319-322: the r2 values and stdev values are listed, but nothing about their implications is mentioned or explained. It would be good to include one sentence as to what these values are telling the reader.

Section 6: it would help the reader follow the comparisons if the section were better organized, either by city or compound class. There is a lot of back and forth and it makes it rather confusing and more difficult than necessary to follow, as I think this is a very interesting and relevant part of the paper.

Section 7.1: can you clarify which method used to calculate the ERs was used in this comparison with EI? I realize the agreement between the methods was quite good, but it would still be helpful to mention if it was the LRF or PA method.

L390-393/Figure 8: to state that more reactive VOCs are reasonably well represented is difficult to determine from this plot, as some of the compounds are labeled but others are not. I think I would try to be a bit more specific about this and avoid blanket statements. Also because looking at these figures I'm not sure that there is a clear pattern, and it seems that some are just better than others, and it would be much better if this were just discussed more explicitly in the text rather than trying to draw very general

conclusions and then listing a bunch of exceptions. Maybe it would be worth considering different shapes for the different species, or more labeling, or possibly color coding based on reactivity?

L393/L402/L405/L467: in a number of cases the authors state that species are 'reasonably represented' or 'reasonably underestimated'. This seems rather subjective, also because 'reasonably represented' seems to be agreement by within a factor of 2. I would suggest either trying to avoid such subjective statements, or defining somewhere that 'reasonable' is somehow tied to this specific range. At the end of L406, I would suggest to not leave it with reasonably represented, but go further to state that these can still use improvement. (I leave that to the author's discretion.)

L402: the authors state that xylenes are reasonably underestimated, but it seems from the figure that some of the dots for xylenes are as far out as the benzene dots. Please explain what this means or ideally be more specific, as mentioned in the above comment.

L397-406: it would help to explain this method a bit more clearly. Are the ratios always [beirut]/[accmip]? Include this.

L418-420: If this sentence is referring to Figure 10, it seems that a number of the EMEP dots are beyond the 2:1 line for overestimation and even one beyond for underestimation, which is not what the sentence states.

L416-422: generally this section seems to draw conclusions that are far too generalized and not really supported by the figures when one examines the details. A more explicit discussion is warranted to avoid blanket statements.

Conclusions: given many of the above comments requesting more explicit discussion and less generalizations, please make sure that the conclusions text is updated to reflect the revised discussion.

L473-474/L476-477: yes! This is a very important point. Thank you for including it.

Specific edits:

L4: write out volatile organic compounds first followed by the VOCs abbreviation in parentheses.

L36-39: This sentence, 'Future projections...' seems redundant with the previous sentence. Please correct.

L60: '...the road transport sector by the Middle East...' the 'by' should be 'in'

L64: suggest to change this sentence for language to 'The same picture is presented by emission inventories.'

L71 & L73: no 's' after RCP

L71: Replace 'On the opposite' with 'In contrast'

L72: replace 'reaching respectively' with 'with totals of' and move 'respectively' to the end of the sentence.

L80: '...and even more so in cities..' add the 'so'

L111: change '..region at a whole...' to '...region as a whole..'

L112: change 'strong' to 'significant'

L136-137: suggest to change the sentence to read '...proposed by Salameh et al. (2014) by samplinng close to emission sources in real-world operating conditions as far as possible.'

L142: remove 'ambient'

L146-147: suggest to change the sentence to '...by a forested pine and high density residential area.'

L176: change 'each sectoral layer' to 'each sector'

L182: remove 'only'

L212: would be good to include/mention that the tracer was acetylene. (again, not inert, just less reactive)

L224: change 'neglected' to 'less'

L235-236: aromatics and alkanes should be singular

L250: 'on' should be replaced by 'in' before LRF

L328-329: please be clearer whether the more than 80% of the species was true for both seasons or just one season was sufficient to be included.

L371-372: this sentence does not make sense. Differences compared to what? Mecca? Also, 'opposite' should be 'in comparison' or 'in contrast' to.

L394: replace 'remarkable' with 'significant' or something else more appropriate.

L403: replace 'lay around' with 'are closer to'

L429: replace 'homogeneity' with 'agreement'

L436: I would suggest that 'poorer in alkenes' be 'poorer in some alkenes'

L439: suggest to end the sentence after 'MEA', and then revising the next sentence to 'This could be indicative for other MEA countries where emissions data and measurements are scarce.'

Table 2: at the very end of the table caption, please add '..for the VOC/acetylene ERs'

Table 2: at the end of the table, 'nd: not determined' is included but does not show up in the table.

Table 3: in the table caption, the parentheses need to be fixed.

Figure 2/3: having the middle thickness line as a dashed line would help readability

Figure 7: swapping the summer Beijing plot with the summer Mecca plot would put the two Beijing plots next to each other, which is more logical for the reader.
Figure 9: given that the plots seem to be organized by compound class, it would be good to include these compound class labels in the top left corner of the plots for orientation. I also assume that the 'all data' in grey dots is also just for those compound classes. But maybe this is not the case?

―――――――――――――――――――

---

## Referee Comment (RC2) · Anonymous Referee #2 · 19 Sep 2016

This paper describes NMVOCs measurements conducted in summer and winter time in Beirut and the assessment of emission ratios using two different methods. Emission ratios derived from measurements conducted at a suburban site were compared with those performed near a road transport sector. This comparison suggested that for the majority of NMVOCs, road transport is the main source of emission in Beirut. Therefore, ERs derived from the observations at a suburban site were also compared with those extracted from emission inventories for all the anthropogenic sectors. The authors concluded that the overall speciation of anthropogenic sources is reasonably represented. Additionally, ERs derived from near-road measurements were compared

with those from emission inventories only from the road transport, concluding that there is a consistency of ER from road transport within a factor of 2 with EMEP but a high underestimation was found for xylenes and toluene by ACCMIP.

General comments: This paper presents a rich data-set for a region where emission data and measurements are still scarce. The paper is well written and structured and the methodology adapted for deriving ERs and comparing them to emission inventories is rigorous. I recommend the manuscript for publication with minor corrections.

Some minor comments:

I would recommend to shorten the abstract

L170 Rephrase "Only the anthropogenic part of the emissions regarding road transport (SNAP 07) ". As formulated it seems that there is a biogenic part of the road transport

L283: Did you use the same value of OH concentration for both seasons? (5.106 299 molecules.cm-3 ). Could you briefly discuss the impact on the results if this value is different between summer and wintertime?

L222: Please define kOH the first time it appears in the text. Specify that it's a reaction rate coefficient

L225-226: This sentence is not very clear and should be reformulated. What is this threshold value of 8.52x10-12 cm3 molecule-1 s-1?

L270-273: It is hard to believe that there is " no photochemistry during daytime in summer" especially in the Middle east region

L299: Please indicate that the value in the parenthesis is the assumed concentration of OH.

L306 Replace "keeps" with "remains" or "is"

L337: the sentence should be completed in order to specify that wintertime ER agree

at +/- 30% . . . with the summertime ERs

Figure 4: Please use different markers for the different compounds

Figure 5. It should be added in the legend that the figure represents only ethane data.

———————————————————

---

## Author Comment (AC1) · 11 Nov 2016

Dear referee,

We would like to thank you for your comments, which significantly improved the quality of this work. We have revised the manuscript entitled "Composition of gaseous organic carbon during ECOCEM in Beirut, Lebanon: new observational constraints for VOC anthropogenic emission evaluation in the Middle East" according to your comments. Kindly find below our response to the comments.

Sincerely yours,

Thérèse Salameh

**Anonymous Referee #1**

This paper provides a nice presentation of emission ratios (ER) calculated by two different methods, and then follows up with a comparison with regional to global emission inventories. This paper is especially relevant given the lack of measurement data from this region of the world. Overall, the paper is well organized, and quite relevant. I would however like to see a few revisions before it is accepted for publication. More general comments and specific edits/suggestions, including some language corrections, follow.

**General comments:**

1. In general, the paper cites a lot of work from authors, which is fine given the lack of work in this region on the topic, but it would be good if there could also be some more citations that are beyond self-citation. For example, on L238, that toluene and xylenes are related to traffic sources, there are references other than Salameh 2016 that are earlier and more authoritative on this, as this is something that is more broadly true and not limited to the Middle East. It would be good to integrate some of these as well.

*We referred to Salameh et al. 2016 because it is a source apportionment study where these species are present in the road-transport profile. Nevertheless, we added two other references for other regions: Watson et al. 2001 and Badol et al. 2008.*

2. L95: CO is not inert. It is much less reactive than many of the VOCs but it is definitely not inert.

*At the time scale of air mass transport in Beirut (hour), CO can be considered as inert. The word "inert" is removed from the whole manuscript. Thank you.*

3. P7: this text discusses where speciations in emission inventories come from, but it does not make clear which species/speciation information is used from which sources for the comparisons in the study. Please try to be more explicit/clearer about this, since a variety of inventories, etc are discussed.

*We have indicated in line 191 that "we rely on the ACCMIP inventory from ECCAD database which provides a detailed speciation of VOCs including acetylene." In line 201 we mentioned that the "speciation of NMVOC emissions is performed using the RETRO inventory" and in line 187 we mentioned that the same methodology for the NMVOCs speciation is used in MACCity and ACCMIP emission inventories. In line 207, we specified that "the emission fluxes of 11 individual NMVOCs as well as CO" were extracted from the ECCAD database. The list of target species is reported in Table 3.*

4. L202-204: This sentence does not make sense. Why does what is said above make the approach for Beirut valid? Please explain this more clearly.

*By this, we mean that the use of emission ratios from our measurement site is relevant because there is no significant spatial variability of the fluxes of the grids covering Lebanon and the site is representative of Beirut anthropogenic emissions.*
*The sentence is changed to: "Since the flux of the 8 grids is spatially homogeneous for all compounds, we calculated a mean flux for each NMVOC to derive the emission ratio from the inventory. Comparing emission ratios from the inventory to observations collected at one measurement site stays relevant. Indeed the sub-urban site is far enough from strong direct emissions (industrial, road transport) since they can hide the emissions from distant sources, and it receives air masses coming from Greater Beirut Area which includes the city of Beirut and close suburbs."*

5. L252: I find it very hard to believe that photochemical removal/photochemistry is truly 'negligible' in winter.

*The impact of photochemistry was largely described in Salameh et al. 2015 as follows: It was assessed through the comparison of night-time and daytime scatterplots during summer and during winter. We assume that there is no photochemistry during night-time and the composition of emissions does not change. The advantage of using the mixing ratios of pairs of ambient NMHC species is that they are not sensitive to dilution and air-mass mixing compared with absolute concentrations themselves. Examining the ratios is useful in exploring the influence of photochemical depletion for compounds with different atmospheric lifetimes. The results show that the most reactive compounds are slightly affected by photochemistry (ethylene, m,p-xylenes) in summer and not affected at all in winter. As for less-reactive NMHC, there is no effect of photochemistry during both seasons. Please refer to Salameh et al. 2015 for more detailed information.*

6. L270-274: Here the authors state that ER were derived using all the data 'since there is no effect of photochemistry even during the day in summer'. That simply cannot be true. No photochemistry during summer mid-day? If the authors wanted to argue that the LRF were derived based on all data because the emissions were fresh and therefore had not undergone significant photochemistry I could follow the argument/logic, but to state no photochemistry seems very unreasonable. Also, in section 4 the degree of chemical removal is stated to be kOH dependent. Overall, the discussion in section 4 and section 5 should be made more consistent, but if 'no photochemistry' rather than e.g., 'fresh emissions' wants to be argued, the data to prove this will need to be much more substantial. That there is substantial influence from emissions is discussed and a relevant point.

*Kindly refer to the previous comment regarding the impact of photochemistry during summer and winter. Additionally, this sentence from L. 285 to 289 concerns only the less reactive species as we have stated in the beginning of the sentence "for other species". The more reactive species affected by photochemical removal during the midday in summer are treated in a different way (please see lines 277 – 282).*

*Furthermore, the impact of fresh emissions was addressed many times: in lines 239-240 "…compensation of chemical removal at midday by evaporative emission cannot be excluded for aromatics like toluene" and lines 250 – 252 "…strong enrichment of aromatics during the day originating from the traffic related source where toluene and m,p-xylenes are significantly present."We added "fresh" to the line 239: "…at midday by fresh evaporative emission".*

7. Section 5.2: It would be helpful to bring Figure 4 into the discussion earlier. I think visualizing this and referring to the plot will help the explanation. It would also be helpful if a number of points could be more explicitly mentioned. The ER is determined by extrapolating the photochemical age to t=0, so that means the y-intercept is the ER, correct? State this explicitly. Photochemical age is being shown on the x-axis, but what does that correspond to in terms of the equation that is shown? It would be good to be explicit about this too.

*The section 5.2 is changed according to the comments. We introduced figure 4 in the first paragraph: "…NMVOC species is dominated by a reaction with the OH radical. In this method, the ratio of NMVOC with acetylene is plotted versus the photochemical age as shown in Figure 4." We added the explanation to line 329: "…The emission ratios of NMVOCs are estimated by extrapolating the photochemical age to zero which is the intercept on the y-axis of the linear fit (figure 4)." We added "Δt" in the caption of figure 4.*

8. L319-322: the r2 values and stdev values are listed, but nothing about their implications is mentioned or explained. It would be good to include one sentence as to what these values are telling the reader.

*The sentence is changed to: "The calculated determination coefficients $R^2$ with acetylene and CO ranged from 0.3 for some species like ethane to 0.9 for the majority of the species in winter, and with acetylene in summer from 0.5 to 0.9 showing the importance of combustion related sources during both seasons. The standard deviation of the ER determined by the photochemical method was low and varied between zero and 0.04 for propane, the coefficient of variation was below 3% for the majority of the species indicating the robustness of the photochemical method."*

9. Section 6: it would help the reader follow the comparisons if the section were better organized, either by city or compound class. There is a lot of back and forth and it makes it rather confusing and more difficult than necessary to follow, as I think this is a very interesting and relevant part of the paper.

*The section 6 is modified as follows: "...Usually ERs agree within a factor of 2 except for aromatics (benzene excepted) and some alkanes (C2 – C5). Those species are related to the unburned fuel fraction and natural gas or liquefied petrol usage.*
*Among C2-C5 alkanes and regardless of the season, ER of ethane is much lower in Beirut than Los Angeles, Beijing, and Strasbourg but similar to Saudi Arabia since the natural gas source is not widely used in Lebanon and Middle East countries (Salameh et al., 2015). Regardless of the season, ERs of aromatics are higher in Beirut compared to northern post-industrialized countries and even the Middle Eastern city Mecca. One should note that aromatic differences are quite significant between the two Middle Eastern cities, from a factor of 3 up to a factor of 6 for (m,p)-xylenes. The maximum difference is observed between Beirut and Strasbourg and reached a factor of 10 for m,p-xylenes in winter. Differences are greater in winter than in summer as a consequence of a marked seasonal variability of ER in other cities (Strasbourg and Beijing) in contrast to Beirut. In Beirut, the aromatics are emitted from combustion related sources and from gasoline evaporation which accounts for more than 40% in winter as well as in summer (Salameh et al., 2016). ERs of alkenes, which are combustion products, usually agree within a factor of two between Beirut, Los Angeles, Beijing, and Strasbourg in both seasons, except for C4 and C5 alkenes with LA and Beijing; whereas they are higher in Mecca due to their additional evaporative origin (Simpson et al., 2014)."*

10. Section 7.1: can you clarify which method used to calculate the ERs was used in this comparison with EI? I realize the agreement between the methods was quite good, but it would still be helpful to mention if it was the LRF or PA method.

*The method used is LRF method and it was added to the sentence line 419: "...ratios during summer and winter obtained by the LRF method."*

11. L390-393/Figure 8: to state that more reactive VOCs are reasonably well represented is difficult to determine from this plot, as some of the compounds are labeled but others are not. I think I would try to be a bit more specific about this and avoid blanket statements. Also because looking at these figures I'm not sure that there is a clear pattern, and it seems that some are just better than others, and it would be much better if this were just discussed more explicitly in the text rather than trying to draw very general conclusions and then listing a bunch of exceptions. Maybe it would be worth considering different shapes for the different species, or more labeling, or possibly color coding based on reactivity?

*The sentence is changed to: "Except benzene, xylenes to a less extent and long-lived alkanes, ER relative to acetylene agree within a factor of 2 between observations and inventory suggesting that the VOC speciation in ACCMIP is better represented for more reactive VOCs such as alkenes and some aromatics." The color coding is based on NMVOC groups but also gives information about the reactivity: Alkanes as less reactive; Alkenes as more reactive; aromatics as more reactive except benzene. When the ERs are < or > 2, the species are labeled.*

12. L393/L402/L405/L467: in a number of cases the authors state that species are 'reasonably represented' or 'reasonably underestimated'. This seems rather subjective, also because

'reasonably represented' seems to be agreement by within a factor of 2. I would suggest either trying to avoid such subjective statements, or defining somewhere that 'reasonable' is somehow tied to this specific range. At the end of L406, I would suggest to not leave it with reasonably represented, but go further to state that these can still use improvement. (I leave that to the author's discretion.)

*Thank you for this comment. We replaced "reasonable" with "better" in lines 424-439-505 and in the abstract (line 21). We removed "reasonable" in line 435. We added a sentence in line 414: "We consider that the speciation is "reasonable" when there is an agreement within a factor of two between observations and emission inventory."*

*We added in line 440: "…SOA precursors in ACCMIP is better represented than the others species but it is still need improvement."*

13. L402: the authors state that xylenes are reasonably underestimated, but it seems from the figure that some of the dots for xylenes are as far out as the benzene dots. Please explain what this means or ideally be more specific, as mentioned in the above comment.

*We removed the word "reasonable" and changed the sentence to: "…xylenes are underestimated in the ACCMIP global emission inventory but not systematically."*

14. L397-406: it would help to explain this method a bit more clearly. Are the ratios always [beirut]/[accmip]? Include this.

*An explanation was added: "In order to consolidate our conclusions regarding VOC speciation within ACCMIP, we performed the systematic calculation of the ratios of every NMVOC to each of the other NMVOCs (NMVOCi) in the global emission inventory and in the observations (Coll et al., 2010) separately, and then we reported in figure 9, the ERs obtained from ACCMIP versus those obtained from the observations."*

15. L418-420: If this sentence is referring to Figure 10, it seems that a number of the EMEP dots are beyond the 2:1 line for overestimation and even one beyond for underestimation, which is not what the sentence states.

*The description is modified to: "ERs of alkenes from the road transport sector are usually consistent within a factor of 2 for the regional emission inventory EMEP and the global inventory ACCMIP. In EMEP, benzene and ethane are overestimated whereas butanes are underestimated; while xylenes and toluene are underestimated over a factor of 2 by ACCMIP. At a more detailed level, by calculating the ratios for individual NMVOC, figure 11 confirms that xylenes and toluene are underestimated species by both inventories, and benzene is overestimated by EMEP."*

*Furthermore, the species with ER< or > 2 are labeled in figure 10.*

16. L416-422: generally this section seems to draw conclusions that are far too generalized and not really supported by the figures when one examines the details. A more explicit discussion is warranted to avoid blanket statements.

*Please refer to the previous comment. Thank you!*

17. Conclusions: given many of the above comments requesting more explicit discussion and less generalizations, please make sure that the conclusions text is updated to reflect the revised discussion.

*We modified the conclusions according to above mentioned comments from line 503 – 510: "This comparison suggests that the overall speciation of anthropogenic sources for major hydrocarbons that act as ozone and SOA precursors in ACCMIP is better represented than other species but it is still need improvement.*

*The road transport ER relative to acetylene derived from near-field measurements are compared to ER from ACCMIP and EMEP regional emission inventory for the road transport sector. ERs of more reactive species (alkenes and aromatics except benzene) are usually consistent within a factor of 2 for the regional emission inventory EMEP while xylenes and toluene are underestimated over a factor of 2 by ACCMIP."*

18. L473-474/L476-477: yes! This is a very important point. Thank you for including it.

*Thank you for all your valuable comments.*

**Specific edits:**

1. L4: write out volatile organic compounds first followed by the VOCs abbreviation in parentheses.

*Changed.*

2. L36-39: This sentence, 'Future projections...' seems redundant with the previous sentence. Please correct.

*The sentence is removed.*

3. L60: '...the road transport sector by the Middle East...' the 'by' should be 'in'

*The sentence is changed to: "They also found that the road transport sector in the Middle East region is a contributor to the global emissions of CO and NOx as significant as road transport in Western Europe and North America."*

4. L64: suggest to change this sentence for language to 'The same picture is presented by emission inventories.'

*The sentence is changed accordingly.*

5. L71 & L73: no 's' after RCP

*The "s" is removed.*

6. L71: Replace 'On the opposite' with 'In contrast'

*Changed.*

7. L72: replace 'reaching respectively' with 'with totals of' and move 'respectively' to the end of the sentence.

*The sentence is changed to: "In contrast, the NMVOC emissions have been strongly decreasing in USA and Europe, with totals of 7 and 10 Tg/year in 2010 respectively according to RCP 8.5 (Figure 1)."*

8. L80: '...and even more so in cities..' add the 'so'

*Added.*

9. L111: change '..region at a whole...' to '...region as a whole..'

*Changed.*

10. L112: change 'strong' to 'significant'

*Changed.*

11. L136-137: suggest to change the sentence to read '...proposed by Salameh et al. (2014) by sampling close to emission sources in real-world operating conditions as far as possible.'

*The sentence is changed accordingly.*

12. L142: remove 'ambient'

*Removed.*

13. L146-147: suggest to change the sentence to '...by a forested pine and high density residential area.'

*The sentence is changed to: "The site is surrounded by a forested pine and high density residential area."*

14. L176: change 'each sectoral layer' to 'each sector'

*Changed.*

15. L182: remove 'only'

*Removed.*

16. L212: would be good to include/mention that the tracer was acetylene. (again, not inert, just less reactive)

*The sentence is changed to: "...the average diurnal normalized profiles to the midnight value of some NMVOC relative to a tracer to examine the relative importance of these processes (Borbon et al., 2013). Acetylene was chosen as a less reactive combustion tracer and its normalized diurnal profile is reported in each panel (grey shaded)."*

17. L224: change 'neglected' to 'less'

*Changed.*

18. L235-236: aromatics and alkanes should be singular

*The "s"is removed.*

19. L250: 'on' should be replaced by 'in' before LRF

*We replaced "on" by "in".*

20. L328-329: please be clearer whether the more than 80% of the species was true for both seasons or just one season was sufficient to be included.

*One season was sufficient to be included, the sentence is changed to: "The ER derived from the observations (summer and winter) are comparable at ±50% to the ratios of road transport sector, during at least one season, for more than 80% of the species."*

21. L371-372: this sentence does not make sense. Differences compared to what? Mecca? Also, 'opposite' should be 'in comparison' or 'in contrast' to.

*The sentence is changed to: "Differences are greater in winter than in summer as a consequence of a marked seasonal variability of ER in other cities (Strasbourg and Beijing) in contrast to Beirut."*

22. L394: replace 'remarkable' with 'significant' or something else more appropriate.

*"Remarkable" is replaced with "significant".*

23. L403: replace 'lay around' with 'are closer to'

*"Lay around" is replaced with "are closer to".*

24. L429: replace 'homogeneity' with 'agreement'

*"Homogeneity" is replaced with "agreement".*

25. L436: I would suggest that 'poorer in alkenes' be 'poorer in some alkenes'

*"Some" is added.*

26. L439: suggest to end the sentence after 'MEA', and then revising the next sentence to 'This could be indicative for other MEA countries where emissions data and measurements are scarce.'

*The sentence is changed accordingly.*

27. Table 2: at the very end of the table caption, please add '..for the VOC/acetylene ERs'

*The caption is changed to: "...Bold characters indicate the similarity at ±50% of the VOC/acetylene ER from the measurement campaigns to the one of near-field measurements."*

28. Table 2: at the end of the table, 'nd: not determined' is included but does not show up in the table.

*"nd: not determined" is removed.*

29. Table 3: in the table caption, the parentheses need to be fixed.

*The parentheses are fixed.*

30. Figure 2/3: having the middle thickness line as a dashed line would help readability

*The figures 2 and 3 are changed.*

31. Figure 7: swapping the summer Beijing plot with the summer Mecca plot would put the two Beijing plots next to each other, which is more logical for the reader.

*We changed the position of the figures accordingly.*

32. Figure 9: given that the plots seem to be organized by compound class, it would be good to include these compound class labels in the top left corner of the plots for orientation. I also assume that the 'all data' in grey dots is also just for those compound classes. But maybe this is not the case?

*The figure is change accordingly. The grey dots are for all compound classes; this explanation is added to the caption of the figure.*

---

## Author Comment (AC2) · 11 Nov 2016

Dear referee,

We would like to thank you for your comments, which significantly improved the quality of this work. We have revised the manuscript entitled "Composition of gaseous organic carbon during ECOCEM in Beirut, Lebanon: new observational constraints for VOC anthropogenic emission evaluation in the Middle East" according to your comments. Kindly find below our response to the comments.

Sincerely yours,

Thérèse Salameh

**Anonymous Referee #2**

This paper describes NMVOCs measurements conducted in summer and winter time in Beirut and the assessment of emission ratios using two different methods. Emission ratios derived from measurements conducted at a suburban site were compared with those performed near a road transport sector. This comparison suggested that for the majority of NMVOCs, road transport is the main source of emission in Beirut. Therefore, ERs derived from the observations at a suburban site were also compared with those extracted from emission inventories for all the anthropogenic sectors. The authors concluded that the overall speciation of anthropogenic sources is reasonably represented. Additionally, ERs derived from near-road measurements were compared with those from emission inventories only from the road transport, concluding that there is a consistency of ER from road transport within a factor of 2 with EMEP but a high underestimation was found for xylenes and toluene by ACCMIP.

**General comments:** This paper presents a rich data-set for a region where emission data and measurements are still scarce. The paper is well written and structured and the methodology adapted for deriving ERs and comparing them to emission inventories is rigorous. I recommend the manuscript for publication with minor corrections.

**Some minor comments:**

1. I would recommend to shorten the abstract

*We tried to shorten the abstract but unfortunately we couldn't remove any important message, therefore the abstract is still relatively long:*

*"The relative importance of Eastern Mediterranean emissions is suspected to be largely underestimated compared to other regions worldwide. Here we use detailed speciated measurements of volatile organic compounds (VOCs) to evaluate the spatial heterogeneity of VOC urban emission composition and the consistency of regional and global emission inventories downscaled to Lebanon (EMEP, ACCMIP, and MACCity). The assessment was conducted through the comparison of the emission ratios extracted from the emission*

*inventories to the ones obtained from the hourly observations collected at a sub-urban site in Beirut, Lebanon during summertime and wintertime ECOCEM campaigns. The observed ERs were calculated by two independent methods. ER values from both methods agree very well and are comparable to the ones of the road transport sector from near-field measurements for more than 80% of the species. There is no significant seasonality in ER for more than 90% of the species unlike the seasonality usually observed in other cities worldwide. Regardless of the season, ERs agree within a factor of 2 between Beirut and other representative worldwide cities except for the unburned fuel fraction and ethane. ERs of aromatics (benzene excepted) are higher in Beirut compared to northern post-industrialized countries and even the Middle Eastern city Mecca. The comparison of the observed ER to the ones extracted from ACCMIP and MACCity global emission inventories suggests that the overall speciation of anthropogenic sources for major hydrocarbons that act as ozone and SOA precursors in ACCMIP is better represented than other species.*

*The comparison of the specific road transport ER relative to acetylene derived from near-field measurements to ER from ACCMIP and EMEP emission inventories for road transport sector showed that ER of more reactive species are usually consistent within a factor of 2 with EMEP while xylenes and toluene are underestimated over a factor of 2 by ACCMIP.*

*The observed heterogeneity of anthropogenic VOC emission composition between Middle Eastern cities can be significant for reactive VOC but is not depicted by global emission inventories. This suggests that systematic and detailed measurements are needed in the Eastern Mediterranean Basin in order to better constrain emission inventory."*

2. L170 Rephrase "Only the anthropogenic part of the emissions regarding road transport (SNAP 07) ". As formulated it seems that there is a biogenic part of the road transport

*The sentence is changed to: "Only the emissions regarding road transport (SNAP 07) is included in this study."*

3. L283: Did you use the same value of OH concentration for both seasons? ($5.10^6$ molecules.cm$^{-3}$). Could you briefly discuss the impact on the results if this value is different between summer and wintertime?

*The photochemical age method was only applied to summertime dataset as indicated in L.292. Additionally, previous studies (Borbon et al. 2013; Warneke et al. 2007) have shown that the emission ratios determined with this method are not affected when reducing or increasing the OH values by a factor of 2. This was mentioned in L.329 – 331.*

4. L222: Please define kOH the first time it appears in the text. Specify that it's a reaction rate coefficient.

*The sentence is changed to: "The degree of chemical removal during the day is $k_{OH}$-dependent ($k_{OH}$: rate coefficient for the reaction with OH)."*

5. L225-226: This sentence is not very clear and should be reformulated. What is this threshold value of $8.52 \times 10^{-12}$ cm$^3$ molecule$^{-1}$ s$^{-1}$?

*The sentence is changed to: "Then the importance of daytime maximum and minimum concentrations becomes modulated by chemical removal when $k_{OH}$ is higher than $8.52 \times 10^{-12}$ $cm^3$ $molecule^{-1}$ $s^{-1}$ (rate coefficient of ethylene with OH)."*

6. L270-273: It is hard to believe that there is "no photochemistry during daytime in summer" especially in the Middle East region

*The impact of photochemistry was largely described in Salameh et al. 2015 as follows: It was assessed through the comparison of night-time and daytime scatterplots during summer and during winter. We assume that there is no photochemistry during night-time and the composition of emissions does not change. The advantage of using the mixing ratios of pairs of ambient NMHC species is that they are not sensitive to dilution and air-mass mixing compared with absolute concentrations themselves. Examining the ratios is useful in exploring the influence of photochemical depletion for compounds with different atmospheric lifetimes. The results show that the most reactive compounds are slightly affected by photochemistry (ethylene, m,p-xylenes) in summer and not affected at all in winter. As for less-reactive NMHC, there is no effect of photochemistry during both seasons. Please refer to Salameh et al. 2015 for more detailed information.*

7. L299: Please indicate that the value in the parenthesis is the assumed concentration of OH.

*The sentence is changed to: "…the reaction of those compounds with OH radical ([OH] = $5.10^6$ $molecules.cm^{-3}$)."*

8. L306 Replace "keeps" with "remains" or "is"

*"Keeps" is replaced with "remains".*

9. L337: the sentence should be completed in order to specify that wintertime ER agree at +/- 30% . . . with the summertime ERs

*The sentence becomes: "The wintertime emission ratios for most NMVOC species agree at ±30% (slope of 0.71) with the summertime ERs and within a factor of 2 and a high determination coefficient of 0.94."*

10. Figure 4: Please use different markers for the different compounds

*Figure 4 is changed.*

11. Figure 5. It should be added in the legend that the figure represents only ethane data.

*The figure represents the ERs of all the compounds; each dot corresponds to a species. We just indicated the name of the compound with ER<2 which is ethane.*